# Prognostic Significance of MRE11 Overexpression in Colorectal Cancer Patients

**DOI:** 10.3390/cancers15092438

**Published:** 2023-04-24

**Authors:** Vincent Ho, Liping Chung, Kate Wilkinson, Vivienne Lea, Stephanie H. Lim, Askar Abubakar, Weng Ng, Mark Lee, Tara L. Roberts, Wei Chua, Cheok Soon Lee

**Affiliations:** 1School of Medicine, Western Sydney University, Sydney, NSW 2560, Australia; liping.chung@westernsydney.edu.au (L.C.); vivienne.lea@health.nsw.gov.au (V.L.); askar.abubakar@westernsydney.edu.au (A.A.); tara.roberts@westernsydney.edu.au (T.L.R.);; 2Ingham Institute for Applied Medical Research, Liverpool, NSW 2170, Australia; kate.wilkinson1@health.nsw.gov.au (K.W.); stephanie.lim@health.nsw.gov.au (S.H.L.); 3Department of Medical Oncology, Liverpool Hospital, Liverpool, NSW 2170, Australia; weng.ng@health.nsw.gov.au; 4Department of Anatomical Pathology, Liverpool Hospital, Liverpool, NSW 2170, Australia; 5Macarthur Cancer Therapy Centre, Campbelltown Hospital, Sydney, NSW 2560, Australia; 6Department of Radiation Oncology, Liverpool Hospital, Liverpool, NSW 2170, Australia; mark.lee2@health.nsw.gov.au; 7South Western Sydney Clinical School, University of New South Wales, Liverpool Hospital, Liverpool, NSW 2170, Australia; 8Discipline of Medical Oncology, School of Medicine, Western Sydney University, Liverpool Hospital, Liverpool, NSW 2170, Australia; 9Discipline of Pathology, School of Medicine, Western Sydney University, Sydney, NSW 2560, Australia

**Keywords:** colorectal cancer (CRC), MRE11, DNA damage response, prognosis, biomarkers, primary tumor site, lymph node involvement

## Abstract

**Simple Summary:**

Colorectal cancer (CRC) is the third-most frequent cancer in the world and the second in terms of mortality rate. Consequently, the identification of biomarkers that can be used to predict CRC prognosis is of considerable interest. Proteins involved in the detection and repair of DNA damage have been implicated in the development, evolution, and response to therapy for many cancers, including CRC. In this study, we investigated the prognostic value of one such factor, known as meiotic recombination 11 (MRE11)—a member of an essential DNA repair complex. We found that elevated MRE11 expression was associated with poor overall and disease-free survival, showing high prognostic value for the subgroup of patients with right-sided primary CRC. Collectively, our findings suggest that MRE11 is an independent biomarker in CRC, which can be leveraged to improve patient outcomes.

**Abstract:**

Meiotic recombination 11 (MRE11) plays a critical role in the DNA damage response and maintenance of genome stability and is associated with the prognosis for numerous malignancies. Here, we explored the clinicopathological significance and prognostic value of MRE11 expression in colorectal cancer (CRC), a leading cause of cancer-related deaths worldwide. Samples from 408 patients who underwent surgery for colon and rectal cancer between 2006 and 2011, including a sub-cohort of 127 (31%) patients treated with adjuvant therapy, were analyzed. In Kaplan–Meier survival analyses, we found that high MRE11 expression in the tumor center (TC) was significantly associated with poor disease-free survival (DFS; *p* = 0.045) and overall survival (OS; *p* = 0.039). Intriguingly, high MRE11 expression in the TC was also significantly correlated with reduced DFS (*p* = 0.005) and OS (*p* = 0.010) in the subgroup with right-sided primary CRC. In multivariate analyses, high MRE11 expression (hazard ratio [HR] = 1.697, 95% confidence interval [CI]: 1.034–2.785; *p* = 0.036) and lymphovascular/perineural invasion (LVI/PNI; HR = 1.922, 95% CI 1.122–3.293; *p* = 0.017) showed significant association with worse OS in patients with right-sided tumors but not those with left-sided tumors. Moreover, in patients with right-sided tumors, high MRE11 was associated with worse OS for those with lymph node involvement (*p* = 0.006) and LVI/PNI (*p* = 0.049). Collectively, our results suggest that MRE11 may serve as an independent prognostic marker in those with right-sided severe CRC, with clinical value in the management of these patients.

## 1. Introduction

Colorectal cancer (CRC) ranks among the top three causes of cancer and cancer-related mortality worldwide, leading to approximately 10% of all cancer deaths [1]. In recent years, the implementation of screening guidelines and the development of improved treatment options have reduced CRC incidence and mortality in older individuals. However, rising rates of CRC in those under 50 and in developing countries associated with the adoption of a western diet have highlighted the need for biomarkers to better determine disease prognosis and predict patient response to chemo/radiotherapy, which can be highly variable [1,2,3]. Accordingly, numerous studies have investigated the prognostic value of various molecular markers in CRC [4,5,6,7,8]. However, only a small number have been adopted for clinical use, and predicting prognosis and response to therapy remains a considerable challenge for patients with this complex and heterogeneous disease [9].

Genomic instability is a hallmark and a driver of many cancers, and consequently, tumors often show mutations in, or aberrant expression of, key genes involved in the detection and repair of DNA damage [10,11]. Such factors can further serve as prognostic indicators for disease severity and progression, a possibility that has been extensively explored by us and others in CRC [12,13,14,15,16,17]. Moreover, given that radiotherapy and chemotherapies induce cell cycle arrest and cell death via the generation of double-strand breaks (DSBs) and other DNA lesions, alterations in damage-sensing and repair proteins can further affect individual patient responses to cancer treatments [9,18,19].

The conserved heterotrimeric complex formed by meiotic recombination 11 (MRE11), DNA repair protein Rad50 (RAD50), and Nijmegen breakage syndrome 1 (NBS1) detects DSBs and acts upstream of several DNA damage response (DDR) pathways [20]. The MRN (Mre11/Rad50/Nbs1) complex binds to and joins DNA ends together, generating single-stranded DNA through the nuclease activity of MRE11 and initiating repair [20]. MRN further recruits and activates ataxia telangiectasia mutated (ATM), a serine/threonine protein kinase in the phosphatidylinositol 3-kinase-related kinase (PIKK) family [21]. ATM, in turn, phosphorylates all three MRN subunits, further activating downstream repair pathways and expression of checkpoint proteins, leading to cell cycle arrest and repair of DNA damage [22]. MRE11 was further shown to be necessary for the processing of topoisomerase II–DNA complexes that arise as part of normal cellular metabolism, potentially protecting from complex-induced cytotoxicity [23]. Moreover, MRE11 is also present in mammalian mitochondria, where it binds to mitochondrial DNA and may protect from damage due to reactive oxygen species (ROS) [24]. It was recently shown that deficiencies in key mismatch repair (MMR) proteins can lead to deregulated mitochondrial metabolism and increased ROS sensitivity [25]. However, it is unclear whether elevated MRE11 may play a protective role in this context.

Given the crucial role of the MRN complex in the maintenance of genome integrity, its constituent proteins and interacting partners have been investigated as potential prognostic indicators for numerous malignancies, including CRC [26]. Pavelitz and colleagues found that loss of MRE11 is associated with improved overall survival (OS) and long-term disease-free survival (DFS) in a small cohort of patients with stage III colon cancer, independent of treatment [27]. Consistent with these findings, we previously performed a combined analysis of MRE11 and ATM expression in a cohort of 262 rectal cancer patients, which revealed that elevated expression of these proteins in the tumor center is a predictive marker for OS and DFS, as well as poor response to neoadjuvant radiotherapy [28]. Similarly, in a separate study, we found that overexpression of the entire MRN complex in rectal cancer correlates with a worse prognosis and poor response to neoadjuvant radiotherapy [29]. Collectively, these findings highlight the potential prognostic value of the MRN proteins in CRC.

Here, to address this possibility and determine whether MRE11 expression alone holds clinicopathological significance and can predict outcomes in CRC patients, we analyzed samples from 408 individuals with colon and rectal cancer, a subset of which were treated with adjuvant therapy.

## 2. Materials and Methods

### 2.1. Patients and Tissue Specimens

Patients were recruited from Liverpool Hospital, Sydney, Australia. Enrollment criteria were histologically confirmed stage I-IV CRC. A total of 489 patients were identified, and 408, for whom tissues were available from the Anatomical Pathology Department, were enrolled in the study. In the course of standard care, all patients were treated by surgical mesorectal excision and anterior or abdominoperineal resection and chemo/radiotherapy. Chemo/radiotherapy consisted of 25 Gy radiation administered in five sessions or treatment with 50.4 Gy administered in 28 sessions, concurrently with 5-fluorouracil (5FU). Patients were followed at 12-month intervals for 150 months. Follow-up included clinic visits, blood tests, colonoscopy, and imaging, as recommended by the treating specialist. This study was approved by the South Western Sydney Local Health District Human Research Ethics Committee (HREC Reference: HREC/14/LPOOL/186; project number 14/103), Sydney, Australia.

### 2.2. Sample Preparation and Tissue Microarrays

Tissue samples were available from five sites: tumor center (TC); tumor periphery at the invasive edge (TP); normal mucosa adjacent to the tumor; normal mucosa distal to the tumor; and involved lymph nodes. Hematoxylin and eosin (H&E) sections were reviewed to localize the most representative areas of tumor and normal colorectal mucosa in tissue samples. Two 1 mm diameter cores were then obtained from pre- and post-operative formalin-fixed, paraffin-embedded (FFPE) tumor samples. Cores were then transferred into pre-drilled wells of tissue microarray blocks using a Beecher Manual Tissue Microarrayer (Beecher Instruments Inc., Sun Prairie, WI, USA), and the blocks were mounted on slides for immunohistochemical analysis.

### 2.3. Immunohistochemistry

Immunohistochemical staining was performed as described [15]. Briefly, slides were incubated with monoclonal anti-MRE11 primary antibody (1:600; Cat. #ab214; Abcam, Cambridge, UK) for 60 min at room temperature. Slides were washed in Tris-buffered saline with Tween-20 (TBS-T) and incubated for 15 min with DAKO EnVision FLEX + Mouse LINKER (Glostrup Municipality, Glostrup, Denmark), rinsed with TBS-T, and incubated for 30 min with an anti-mouse secondary antibody (Dako EnVision FLEX/HRP DM822). Color development was elicited with a mixture of EnvisionTM FLEX DAB + Chromogen DM827 and EnvisionTM FLEX Substrate Buffer DM823 (DAKO). Finally, slides were counterstained with hematoxylin, washed with cold water, and dipped 10 times in Scott’s Bluing solution. Slides were rinsed with cold water, dehydrated, and mounted.

Slides were independently scored by at least two pathologists. MRE11 expression was calculated as the product of the percent of cells stained and the average per-cell staining intensity, as described [26]. The intensity was graded as follows: 0, negative; 1, weak; 2, moderate; or 3, strong. The percent of positive cells was graded on a 0 to 4 scale as: 0 (<5%), 1 (5–25%), 2 (26–50%), 3 (51–75%), or 4 (>75%). The intensity score and the percent positive score were multiplied to obtain a weighted score ranging from 0 to 12. Tumor samples were then categorized with respect to the median of the score range as belonging to a low expression group (score range: 0–5) or a high expression group (score range: 6–12). MMR protein expression was determined based on positive or negative staining for MutL homolog (MLH)1, MutS homolog (MSH)2, MSH6, and post-meiotic segregation (PMS2), irrespective of the proportion of cells stained using the standard protocol of the Anatomical Pathology Laboratory, Liverpool Hospital, Liverpool, Australia.

### 2.4. Statistical Analysis

Statistical analysis was performed using SPSS for Windows v.27.0 (IBM Corporation, Armonk, NY, USA). Survival analysis was performed on the entire cohort. Further subgroup analysis was performed using early tumor stage and low-grade tumor as covariates. MRE11 protein expression in samples from the cancer core and periphery was assessed in univariate and multivariate analyses using Kaplan–Meier curves and Cox’s proportional hazard ratio (HR) survival modeling. Sex, age, tumor-node-metastasis (TNM) stage, differentiation, lymph node (LN) involvement, metastasis stage at diagnosis, LVI/PNI, and adjuvant and neoadjuvant treatments were included as covariates. The statistical significance of results from univariate and multivariate analyses was determined using the Mann–Whitney U test. *p* < 0.05 was considered significant.

## 3. Results

### 3.1. Study Population

The median age of the 408 patients included in this study was 70 years (range: 23–96 years). Most subjects were male (53.7%), and most were stage III or IV (80.1%) at diagnosis. Out of 408 patients, 127 (31.1%) received adjuvant chemo/radiotherapy. Patients were followed for a median period of 69.8 months (range: 0.1–157.6 months). Patient characteristics are summarized in Table 1, based on available clinical information.

### 3.2. Association between MRE11 Expression and Clinicopathological Features and Prognosis

We first investigated the association between MRE11 expression levels and clinicopathological characteristics (Table 2). In Table 2, the Pearson chi-square test was used to check if the results of a cross-tabulation are statistically significant. *p* < 0.05 was considered significant. Given that prior studies have suggested prognostic value for MRE11 expression in combination with other DNA repair proteins, specifically within the TC [28,29], we measured MRE11 expression in both the TC and TP of the samples in our cohort. Areas with the highest mitotic activity in the central region of the cancer were designated as the TC, whereas the most mitotically active areas at the outer invasive zone of the tumor were considered the TP. Representative immunohistochemical staining of high and low MRE11 protein expression, mainly distributed in the nucleus, is shown in Figure 1. There were no significant differences in gender, age, tumor stage, lymph node involvement, metastasis, or LVI/PNI in patients with high vs. low MRE11 protein expression. In Kaplan–Meier analyses, we further found that high MRE11 expression in the TC was significantly associated with worse DFS (*p* = 0.045; Figure 1B) and OS (*p* = 0.039; Figure 1C). In contrast, no significant differences in DFS (*p* = 0.357; Figure 1D) or OS (*p* = 0.304; Figure 1E) were observed in patients with high vs. low MRE11 protein expression in the TP.

Approximately 15% of CRC tumors have deficiencies in the MMR pathway, a feature that affects long-term prognosis and sensitivity to certain chemotherapies and immune checkpoint inhibitors [30,31,32,33,34,35]. MMR deficiency (dMMR) results from the loss of normal MMR gene alleles in tumor cells, leading to an increased somatic mutation rate and accumulation of mutations in microsatellite regions, known as microsatellite instability (MSI), in tumors. Ultimately, this genomic instability can affect the function of other genes, including *MRE11*, which was shown to be disrupted in >60% of CRC with dMMR [36,37,38].

We, therefore, examined the status of the MMR pathway in patient samples by evaluating the association between MMR (MLH1, MSH2, MSH6, and PMS2) protein expression and MRE11 protein expression. Loss of MMR protein expression can be detected by immunohistochemistry using antibodies to the target proteins of interest. In turn, this loss of immunohistochemical staining can be used to determine the identity of the mutated genes, with the absence of nuclear staining within a tumor indicating dMMR status. In this study, MMR staining was available for 87 out of 408 patients. To assess the activity of MMR proteins, we measured protein expression in the TC or peripheral invasive front (TP), as determined by interaction with and destruction of the surrounding tissues. Interestingly, we found that MRE11 expression levels in the TP at the invasive edge were associated with loss of PMS2 expression (*p* = 0.031; Table 2), indicating that germline mutation in *PMS2* is associated with lower levels of MRE11. However, expression of the other MMR proteins was not significantly associated with MRE11 expression in either the TC or the TP (Table 2).

Using univariate Cox regression analysis, we then found that high expression of MRE11 in the TC and several other clinicopathological features were significantly associated with reduced OS (Table 3). Moreover, in multivariate Cox analysis, MRE11 expression (HR = 1.434, 95% confidence interval [CI]: 1.026–2.002; *p* = 0.035), age (HR = 2.314, 95% CI: 1.576–3.396; *p* < 0.001), lymph node stage (HR = 3.379, 95% CI: 2.263–5.047; *p* < 0.001), and adjuvant therapy treatment (HR = 0.288, 95% CI: 0.184–0.451; *p* < 0.001) remained significantly associated with OS (Table 3), implying that those markers together are strongly prognostic for OS in patients with metastatic CRC.

### 3.3. Correlation between MRE11 Expression and Survival Outcomes Based on CRC Location

Our above findings indicate that high MRE11 expression in the TC was significantly associated with worse DFS and OS (Figure 1B,C). Given that left- and right-sided CRCs have been shown in numerous studies to be biologically and clinically distinct [39,40,41,42,43], we next measured the associations between MRE11 expression and OS and DFS in those with left- and right-sided CRC. DFS measures the time until a patient experiences disease recurrence, while OS reflects the length of time a patient survives after the start of treatment, regardless of disease status. In Kaplan–Meier analyses, we found that high expression of MRE11 within right-sided CRC was significantly correlated with worse OS (*p* = 0.001; Figure 2B) and DFS (*p* = 0.005; Figure 2D), whereas associations within left-sided CRC were not significant (OS, *p* = 0.878 and DFS, *p* = 0.856; Figure 2A,C, respectively). This finding suggests that MRE11 expression has potential prognostic value in CRC patients with right-sided tumors.

Using multivariate Cox regression analysis, we further demonstrated that high MRE11 expression (HR = 1.697, 95% CI: 1.034–2.785; *p* = 0.036) and LVI/PNI (HR = 1.922, 95% CI 1.122–3.293; *p* = 0.017) were significantly associated with worse OS in patients with right-sided tumors (Table 4), whereas no significant association was found in those with left-sided tumors. These data suggest that those markers together are strongly prognostic for OS in patients within the right-sided CRC subgroup. In contrast, LN involvement, age at/over 70 years, and adjuvant therapy treatment remained significantly associated with reduced OS for right-sided tumor subgroups (Table 4).

### 3.4. Prognostic Implications of MRE11 Expression in Right-Sided CRC Subgroups

Lastly, we assessed the relationship between MRE11 expression and patient survival within the right-sided CRC subgroup. To this end, we performed Kaplan–Meier survival analyses of DFS and OS in patients with high vs. low MRE11 expression who received adjuvant treatment (combined radio- and chemotherapy treatment). Results show that patients in the high MRE11 expression group showed significantly worse OS (*p* = 0.019) and disease-free (*p* = 0.006) survival compared to those with low MRE11 expression (Figure 3A,B). We then performed similar analyses to assess OS at low vs. high MRE11 expression in patients with and without LN involvement. Interestingly, when patients were grouped in this way, high MRE11 expression was associated with worse OS and DFS only in patients with LN-positive tumors (*n* = 93, *p* = 0.006; Figure 3D and *n* = 70, *p* = 0.011; Appendix A, respectively) but not in those with LN-negative tumors (*n* = 108, *p* = 0.512; Figure 3C). Similar results were observed for LVI/PNI; that is, high MRE11 expression was associated with worse OS in patients with LVI/PNI-positive tumors (*n* = 72, *p* = 0.049; Figure 3F), but no significant association was detected for those with LVI/PNI-negative tumors (*n* = 128, *p* = 0.512; Figure 3E). These results suggest that MRE11 overexpression may contribute to tumor progression in high-risk patients with severe disease.

### 3.5. Clinical Significance of MRE11 in CRC

To validate our findings in this study, we also performed the experiments using RNA-seq databases via The Human Protein Atlas (HPA) (https://www.proteinatlas.org, accessed on 18 April 2023) to validate our results and we compared them with two individual cohorts from The Cancer Genome Atlas (TCGA-COAD, *n* = 438 and TCGA-READ, *n* = 159). The log-rank *p*-value for the Kaplan–Meier plot presented the results from the analysis of the correlation between mRNA expression level and patient survival. In colon cancer cohort (TCGA-COAD) data analysis, the Kaplan–Meier curves revealed that high MRE11 tends to correlate with worse survival probability (*n* = 438, *p* = 0.290; Figure 4A). This colon cancer cohort (TCGA-COAD) includes not only right-sided tumors (right-sided colon: ascending colon, two-thirds of the transverse colon) but also a proportion of left-sided tumors (left-sided colon: one-third of the transverse colon, sigmoid colon, and descending colon). In contrast, for the rectal cancer cohort (TCGA-READ), low MRE11 was significantly associated with poor survival probability (*n* = 159, *p* < 0.001; Figure 4B).

## 4. Discussion

Despite the development of improved therapies and the adoption of more stringent screening guidelines over the past decade, colon and rectal cancers remain one of the most prevalent and deadliest malignancies worldwide [1]. These cancers show a high degree of complexity regarding their molecular features and clinical behavior, underscoring the need for the identification of prognostic biomarkers to better predict patient outcomes. Here, we investigated the prognostic value of MRE11, an essential DDR protein that has been implicated in the development and evolution of numerous cancers, including CRC [26,27,28,29,36,37,44,45,46,47]. In a cohort of 408 CRC patients, we found that elevated MRE11 expression in the TC was significantly associated with poor DFS and OS, specifically for patients with right-sided CRC. Multivariate analyses further showed that both high MRE11 expression and LVI/PNI were significantly associated with worse OS in patients with right-sided tumors. Further, in this subgroup of patients, high MRE11 expression was associated with worse OS for patients with lymph node involvement and LVI/PNI, suggesting its value as a prognostic indicator for patients with right-side localized high-risk CRC.

As a member of the MRN complex, MRE11 plays critical roles as both a sensor of DNA damage and a mediator of downstream responses. Given that genomic instability is a fundamental feature of many cancers, this protein and its interacting partners have been investigated as potential prognostic indicators for CRC and other malignancies, in some cases, with conflicting results [26]. One study that analyzed a small cohort of stage III colon cancer patients reported an association between MRE11 loss and improved survival outcomes [27]. In past studies, we further found that elevated combined expression of MRE11 and ATM or of the entire MRN complex in the TC is associated with a worse prognosis and poor response to neoadjuvant radiotherapy in rectal cancer patients [28,29]. Similarly, elevated MRE11 was also associated with disease progression, poor survival outcomes, and radioresistance in oral and lung cancer patients [44,46]. Our current findings are generally consistent with these observations, suggesting that elevated MRE11 expression may contribute to worse outcomes. In contrast, a separate study by Sheridan et al. did not detect an association between MRE11 and either survival or therapeutic response in CRC [48]. Moreover, we note that in our prior study, MRE11 expression only had prognostic value in combination with ATM [28]. These discrepancies may be due, in part, to the relatively smaller sample sizes in these studies.

At present, it remains unclear how elevated MRE11 may contribute to worse survival outcomes in CRC patients. Given its essential role in DNA repair, it is possible that MRE11 helps cancer cells withstand and survive DNA-damaging assaults from chemoradiotherapy. Accordingly, we previously showed that elevated MRE11 in combination with ATM and MRN complex members was associated with poor outcomes in rectal cancer patients who received neoadjuvant therapy [28,29]. Here, only a small number of patients received neoadjuvant therapy (*n* = 16). However, we found that in right-side localized tumors, increased MRE11 was associated with reduced DFS and OS in patients who received adjuvant combined radio- and chemotherapy. We also note that here, as well as in our prior two studies described above [28,29], high MRE11 expression was associated with poor survival outcomes only for those with LN involvement. This may suggest that the effects of elevated MRE11 expression are more pronounced in certain contexts, such as those with advanced metastatic disease.

The right and left regions of the colorectum originate from separate sources during embryogenesis and differ in several physiological aspects, including their vasculature and lymphatic drainage systems. Accordingly, there is a growing body of evidence that right- and left-sided CRCs are also distinct in regard to their gene expression and mutational profiles, as well as outcomes, disease prognoses, and responses to various treatment regimens [41,49,50,51,52,53]. A recent study has further uncovered a possible association between tumor sidedness and MRE11, reporting that elevated MRE11 expression is associated with favorable survival outcomes for patients with left-sided CRC, as well as in the subset of left-sided CRC patients with microsatellite stability [45]. In contrast, here, we found that elevated MRE11 expression is associated with worse outcomes, specifically for CRC patients with right-side-localized CRC. Moreover, we did not detect an association between MRE11 and MMR expression. The reasons for these discrepant results are unclear, although they may result, in part, from cohort-specific or methodological differences. Intriguingly, both studies indicate that the effects of MRE11 overexpression—although opposite from one another—are dependent on tumor location, underscoring the need for future studies aimed at elucidating the precise relationship between MRE11 expression, tumor location, and clinical outcomes in CRC. Furthermore, we were able to validate our results against two individual cohorts from The Cancer Genome Atlas (TCGA). The results revealed that for both right- and left-sided cancers in the TCGA cohort, a high MRE11 expression is associated with a trend towards worse survival, but contrasting data from rectal cancers showed that low MRE11 expression confers worse outcomes. This implies that right-sided cancers by themselves have worse outcomes with high MRE11 expression, which is in keeping with our findings.

A primary strength of this study is our relatively large study population, comprising both colon and rectal cancer patients. This feature allowed us to uncover a robust and specific association between elevated MRE11 expression and worse survival outcomes in individuals with right-sided localized primary CRC. However, immunohistochemical staining for MMR proteins was only available for 87 out of the 408 subjects in our study, making it difficult to interpret our findings, which do not show an association between MRE11 expression and MMR protein expression. Prior studies have reported that dMMR can induce *MRE11* mutations, leading to the formation of a truncated, nonfunction protein [36,37,38]. In turn, there is evidence that MRE11 may play a role in the MMR pathway, and loss of this protein is associated with increased MSI in dMMR CRC [54]. Increased MRN expression was also associated with lower tumor grade and improved prognosis for CRCs with functional MMR [55]. Thus, future studies are needed to better understand the prognostic relationship between the MMR pathway and MRE11 expression in the context of CRC.

## 5. Conclusions

In summary, immunohistochemical staining of tumor samples from 408 individuals with colon and rectal cancer followed by Cox regression analysis revealed that high MRE11 expression in the TC was significantly associated with poor DFS and OS for the overall cohort, as well as for the sub-cohort of patients with right-sided primary CRC. In multivariate analyses, both high MRE11 expression and LVI/PNI showed significant association with worse OS in patients with right-sided tumors but not those with left-sided tumors. Moreover, in patients with right-sided tumors, high MRE11 was associated with worse OS for those with lymph node involvement and LVI/PNI. Collectively, these findings suggest that MRE11 expression may represent an independent prognostic marker for patients with right-sided severe CRC, which holds potential clinical value for improving the care and management of these patients.

## Figures and Tables

**Figure 1 cancers-15-02438-f001:**
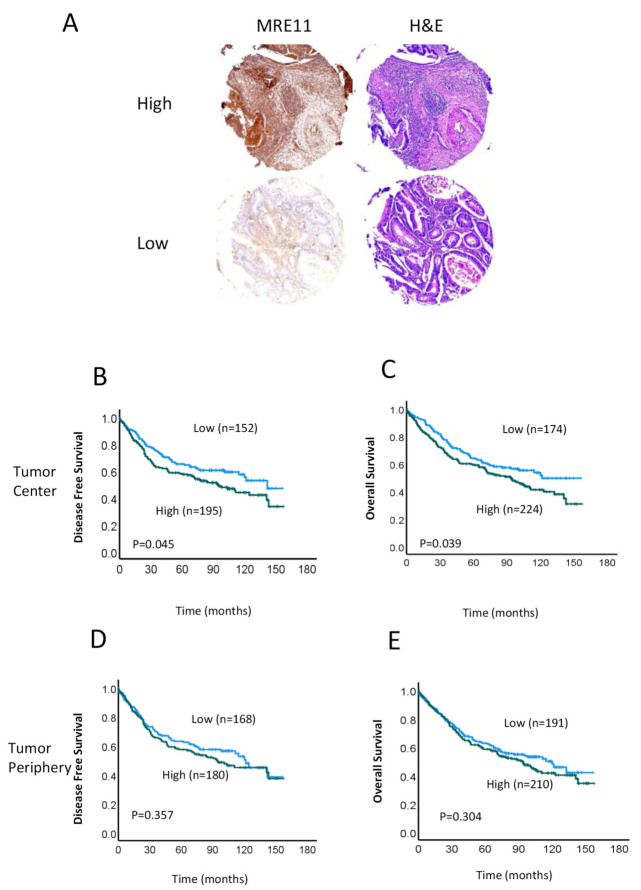
MRE11 expression in colorectal cancer (CRC) tissue samples. (**A**) Representative examples of typical nuclear staining of MRE11 in tumor cells, scored as high or low expression, and corresponding hematoxylin and eosin (H&E) staining are shown. Images were taken at 10× magnification. (40× magnification; scale bar, 50 μm). (**B**–**E**) Kaplan–Meier analyses of disease-free survival (DFS) (**B**,**D**) and overall survival (OS) (**C**,**E**) for patients with high (green line) and low (blue line) MRE11 protein expression in the tumor center (**B**,**C**) and tumor periphery (**D**,**E**).

**Figure 2 cancers-15-02438-f002:**
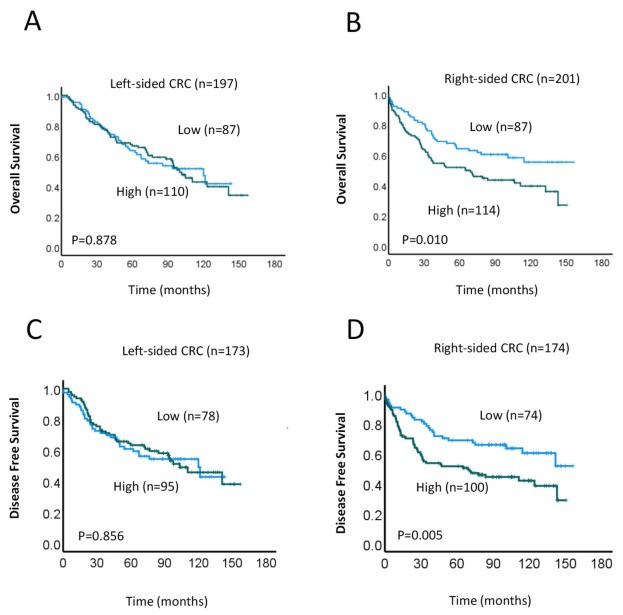
Relationship between MRE11 expression in the TC and survival, based on location of the primary CRC. (**A**–**D**) Kaplan–Meier analysis of OS (**A**,**B**) and DFS (**C**,**D**) in patients with high (green line) or low (blue line) MRE11 expression in left-sided (**A**,**C**) and right-sided (**B**,**D**) colorectal cancers.

**Figure 3 cancers-15-02438-f003:**
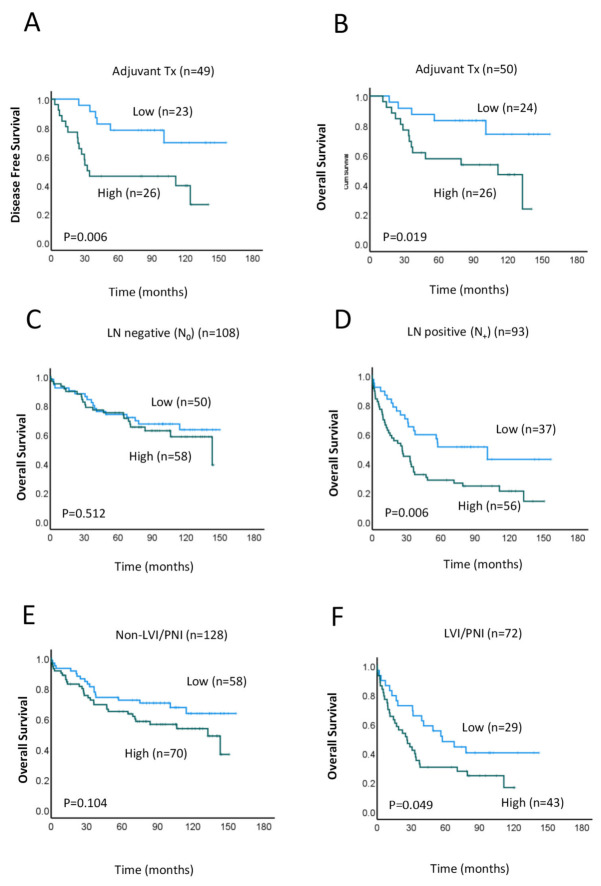
Association between MRE11 expression and prognosis in right-sided CRC. (**A**,**B**) Kaplan–Meier survival analyses of DFS (**A**) and OS (**B**) in patients with high (green line) or low (blue line) MRE11 expression treated with adjuvant chemo/radiotherapy. (**C**–**F**) Kaplan–Meier curves of OS in patients with high (green line) or low (blue line) MRE11 expression in lymph node (LN)-negative (**C**), LN-positive (**D**) tumors, lymphovascular and perineural invasion (LVI/PNI)-negative (**E**) and LVI/PNI-positive (**F**) tumors.

**Figure 4 cancers-15-02438-f004:**
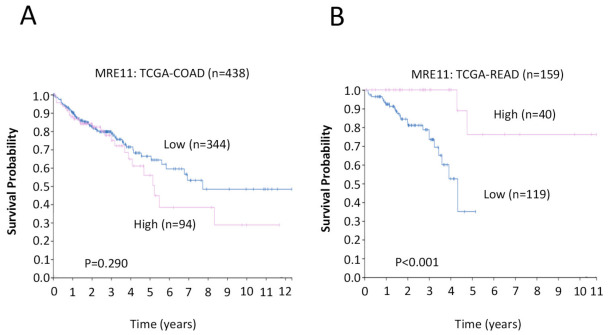
Kaplan–Meier survival analyses comparing survival probability in patients with low MRE11 and high MRE11. Differences between the groups were analyzed by the log-rank test, and *p*-values are shown. (**A**) Survival analyses for colon cancer cohort, and (**B**) for rectal cancer cohort comparing survival probabilities in patients with high (pink line) or low (blue line) MRE11. The cases were provided by The Cancer Genome Atlas (TCGA-COAD and TCGA-READ).

**Table 1 cancers-15-02438-t001:** Patient characteristics for subjects included in this study.

Characteristics	All Patients (*n* = 408)	%
Age median	70 (23–96 years)	
<70	190	46.6
≥70	218	53.4
Sex		
Male	219	53.7
Female	189	46.3
Tumor stage		
T1, T2	81	19.9
T3, T4	327	80.1
Node stage		
N0	213	52.2
N1, N2	195	47.8
Metastasis stage		
M0	353	86.5
M1	55	13.5
Differentiation		
Well/moderate	337	82.5
Poor	71	17.5
LVI/PNI		
Absent	265	65
Present	143	35
Primary tumor site (Right/Left)		
Right	209	51.2
Left	199	48.8
Treatment		
Neoadjuvant therapy		
No	392	96.1
Yes	16	3.9
Adjuvant therapy		
No	281	68.9
Yes	127	31.1
MMR deficiency		
Total	87	21.3 (87/408)
No	64	73.6
Yes	23	26.4

**Abbreviations**: LVI/PNI, lymphovascular and perineural invasion; MMR, mismatch repair.

**Table 2 cancers-15-02438-t002:** Associations between MRE11 expression in the tumor center and tumor periphery and clinicohistopathological data.

		Tumor Centre			Tumor Periphery		
		Low (%)	High (%)	*p*-Value	Low (%)	High (%)	*p*-Value
Sex	Male	53.4	54.0	0.920	54.5	51.9	0.618
	Female	46.6	46.0		45.5	48.1	
Age	<70	44.8	47.3	0.685	46.1	44.3	0.719
	≥70	55.2	52.7		53.9	55.7	
Tumor stage	T1–2	24.1	30.0	0.172	20.9	21.0	0.998
	T3–4	75.9	70.0		79.1	79.0	
Node stage	Negative	55.7	49.6	0.227	51.8	51.9	0.988
	Positive	44.3	50.4		48.2	48.1	
Metastasis stage	M0	87.4	87.1	0.929	88.0	85.7	0.508
	M1	12.6	12.9		12.0	14.3	
Differentiation	Well/moderate	83.3	82.1	0.756	82.2	82.4	0.962
	Poor	16.7	17.9		17.8	17.6	
LVI/PNI	Absent	67.8	62.8	0.297	63.2	66.2	0.562
	Present	32.2	37.2		36.8	33.8	
Adjuvant therapy	No	62.1	65.3	0.535	63.3	65.2	0.714
	Yes	37.9	34.7		36.7	34.8	
Neoadjuvant therapy	No	95.4	96.4	0.605	94.8	97.6	0.132
	Yes	4.6	3.6		5.2	2.4	
MLH1 ^a^	Normal IHC	97.6	97.8	0.961	100	97.1	0.209
	Loss of staining	2.4	2.2		0	2.9	
MSH2	Normal IHC	95.2	97.7	0.529	98.1	97	0.732
	Loss of staining	4.8	2.3		1.9	3	
MSH6	Normal IHC	71.4	84.1	0.157	73.6	84.8	0.221
	Loss of staining	28.6	15.9		26.4	15.2	
PMS2	Normal IHC	76.9	88.1	0.184	75	93.8	0.031
	Loss of staining	23.1	11.9		25	6.2	
dMMR ^b^	No	64.3	80	0.101	69.8	79.4	0.322
	Yes	35.7	20		30.2	20.6	

^a^ Total available data for MMR protein staining (*n* = 87); ^b^ MMR deficiency. **Abbreviations**: IHC, immunohistochemistry; LVI/PNI, lymphovascular and perineural invasion; MMR, mismatch repair.

**Table 3 cancers-15-02438-t003:** Cox regression analyses of MRE11 expression and clinicopathological features with OS.

		Univariate			Multivariate		
	*n (%*)	HR	95% CI	*p*-Value	HR	95% CI	*p*-Value
MRE11 TC ^a^							
High	54.9	1.352	1.015–1.801	0.040	1.433	1.026–2.002	0.035
Low	45.1						
Sex							
Male	53.7	1.19	0.905–1.565	0.214	0.912	0.651–1.267	0.584
Female	46.3						
Age							
<70	46.6	1.828	1.375–2.401	<0.001	2.314	1.576–3.396	<0.001
≥70	53.4						
Tumor stage							
T1, T2	19.9	2.103	1.393–3.173	<0.001	1.292	0.807–2.070	0.286
T3, T4	80.1						
Node stage							
Negative	52.2	2.286	1.727–3.027	<0.001	3.379	2.263–5.047	<0.001
Positive	47.8						
Metastasis stage							
M0	86.5	5.831	4.173–8.150	<0.001	2.747	0.370–20.367	0.323
M1	13.5						
Differentiation							
Well/moderate	82.5	1.564	1.119–2.186	0.009	1.343	0.648–2.782	0.427
Poor	17.5						
LVI/PNI							
Absent	65	1.965	1.488–2.594	<0.001	1.339	1.314–4.075	0.112
Present	35						
Adjuvant therapy							
No	68.9	0.603	0.404–0.806	0.005	0.288	0.184–0.451	<0.001
Yes	31.1						
Neoadjuvant therapy							
No	96.1	1.049	0.538–2.048	0.888	2.207	0.996–4.888	0.151
Yes	3.9						

^a^ MRE11 expression in the tumor center. **Abbreviations**: CI, confidence interval; CRC, colorectal cancer; HR, hazard ratio; LVI/PNI, lymphovascular and perineural invasion; OS, overall survival; TC, tumor center.

**Table 4 cancers-15-02438-t004:** Multivariate analyses of MRE11 expression and clinicopathological features with OS, based on location of the right-sided CRC tumor.

	Right-Sided CRC		
	HR	95% CI	*p*-Value
MRE11			
High	1.697	1.034–2.785	0.036
Low			
Sex			
Male	0.882	0.546–1.417	0.598
Female			
Age			
<70	2.114	1.213–3.685	<0.001
≥70			
Tumor stage			
T1, T2	0.976	0.487–1.955	0.944
T3, T4			
Node stage			
Negative	3.876	2.237–6.716	<0.001
Positive			
Metastasis stage			
M0	6.65	0.786–16.95	0.082
M1			
Differentiation			
Well/moderate	1.479	0.848–2.581	0.168
Poor			
LVI/PNI			
Absent	1.922	1.122–3.293	0.017
Present			
Adjuvant therapy			
No	0.292	0.154–0.565	<0.001
Yes			

**Abbreviations**: CI, confidence interval; CRC, colorectal cancer; HR, hazard ratio; LVI/PNI, lymphovascular and perineural invasion; OS, overall survival.

## Data Availability

The data presented in this study are available on request from the corresponding author. The data are not publicly available due to data size and privacy.

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
