# Peer review of "Prognostic Significance of MRE11 Overexpression in Colorectal Cancer Patients"

_cancers, 2023, doi:10.3390/cancers15092438_

Round 1

Reviewer 1 Report

In this manuscript, the authors analyzed the protein expression of MRE11 in a cohort of colorectal cancer by IHC and investigated their correlations with clinicopathological parameters and patient survival, as well as in a subset of which were treated with adjuvant therapy. In conclusion, they find that MRE11 may serve as an independent prognostic marker in those with right-sided severe CRC, with clinical value in the management of these patients. However, the current manuscript have some logic defects, and the presentation of results show some errors. The manuscript is also not carefully prepared. All these should be revised before the manuscript become to be publishable. The following are some comments for the manuscript in detail. 

1.     In background part, the author state that MRE11 is important for DNA repair, and DNA repair protect cells from genome instability, which is a driver of many cancers. Therefore, high MRE11 expression should promote DNA repair and genome stability, leading to the inhibition of tumorigenesis. However, the results showed that high MRE11 expression was significantly associated with worse survival. This contradiction was not explained in the manuscript. Meanwhile, the patient with chemo/radiotherapy is another story. The treatment induces cell death through severe DNA damage. High MRE11 promote the DNA repair may block the DNA damage-induced cell death. Thus the results that high MRE11 expression was associated with worse survival in patient with adjuvant therapy is consistent with this hypothesis. But their ratio is too low to change the result of large cohort.    

2.     As MRE11 formed complex with Rad50 and Nbs1, it is hard to understand why the author chose to analyze MRE11 alone only. As they also analyze MMR proteins, why did they not analyze the Rad50 and Nbs1?

3.     In fig.1A, the image of H&E stainingis not corresponding to images of either the high MRE11 staining or the low MRE11 staining. The authors should present the H&E staining of the same regions with both high and low MRE11 staining images here.

4.     The results of DFS and OS have different meanings and suggestion to the mechanism. The authors should discuss it.

5.     The method of statistical analysis for p-value in table 2 is not mentioned.

6.     A subtitle should be put between line 164 and 165.

7.     In line 83, “MRN (MRe11, Rad50, Nbs1)” should be “MRN (MRe11/Rad50/Nbs1)” or “MRN (MRe11-Rad50-Nbs1)”.   

Author Response

We thank the reviewer for the constructive comments. A detailed point by point response is provided. 

Reviewer 2 Report

Ho et al. investigated the prognostic potential of MRE11 protein expression in colorectal cancer using over 400 clinical samples. Although the manuscript is well put together and easy for the readers to follow, it lacks novelty and significance. The prognostic potential of MRE11 has been shown in the past and while the authors confirm the findings of the original study, since the cut-offs that were used in this paper differs from the original, it is difficult to assess the reproducibility of the data even this study used an independent cohort. Unfortunately, the protein investigated by the author is not novel and the statistical significance of some of the prognostic measurements were barely significant, despite authors using a significantly large cohort. It would be of benefit of the paper, if authors provided additional evidence that provides support for the clinical significance of MRE11 in colorectal cancer, in order to claim that MRE11 is an independent biomarker.

Specific comments

Please confirm the clinical significance of MRE11 using previously published data sets, if possible, using the same criteria for high and low MRE tumors to validate findings in this paper. One of the major limitations of this study is a lack of evidence for reproducibility. 

Alternatively, if you are unable to find accessible protein data, please use publicly available RNA-Seq data to examine the clinical significance of MRE11. Since this should be relatively straight forward, please use several large cohorts.

Author Response

(The authors gave the same response as above.)

Round 2

Reviewer 1 Report

The authors have addressed all of my concerns, and I have no further points.

Reviewer 2 Report

Authors have adequately addressed the comments.